# Inhibition of the Eukaryotic 80S Ribosome as a Potential Anticancer Therapy: A Structural Perspective

**DOI:** 10.3390/cancers13174392

**Published:** 2021-08-31

**Authors:** Simone Pellegrino, Salvatore Terrosu, Gulnara Yusupova, Marat Yusupov

**Affiliations:** 1Department of Haematology, Cambridge Institute for Medical Research, University of Cambridge, Cambridge CB2 0XY, UK; 2Institut de Génétique et de Biologie Moléculaire et Cellulaire (IGBMC), INSERM U964, CNRS UMR7104, Université de Strasbourg, 67404 Illkirch, France; terrosus@igbmc.fr (S.T.); gulnara.yusupova@igbmc.fr (G.Y.); 3Institute of Fundamental Medicine and Biology, Kazan Federal University, Kazan 420008, Russia

**Keywords:** ribosome, protein synthesis inhibition, anticancer drugs, X-ray crystallography, cryo-EM, drug development

## Abstract

**Simple Summary:**

Unravelling the molecular basis of ribosomal inhibition by small molecules is crucial to characterise the function of potential anticancer drugs. After approval of the ribosome inhibitor homoharringtonine for treatment of CML, it became clear that acting on the rate of protein synthesis can be a valuable way to prevent indefinite growth of cancers. The present review discusses the state-of-the-art structural knowledge of the binding modes of inhibitors targeting the cytosolic ribosome, with the ambition of providing not only an overview of what has been achieved so far, but to stimulate further investigations to yield more potent and specific anticancer drugs.

**Abstract:**

Protein biosynthesis is a vital process for all kingdoms of life. The ribosome is the massive ribonucleoprotein machinery that reads the genetic code, in the form of messenger RNA (mRNA), to produce proteins. The mechanism of translation is tightly regulated to ensure that cell growth is well sustained. Because of the central role fulfilled by the ribosome, it is not surprising that halting its function can be detrimental and incompatible with life. In bacteria, the ribosome is a major target of inhibitors, as demonstrated by the high number of small molecules identified to bind to it. In eukaryotes, the design of ribosome inhibitors may be used as a therapy to treat cancer cells, which exhibit higher proliferation rates compared to healthy ones. Exciting experimental achievements gathered during the last few years confirmed that the ribosome indeed represents a relevant platform for the development of anticancer drugs. We provide herein an overview of the latest structural data that helped to unveil the molecular bases of inhibition of the eukaryotic ribosome triggered by small molecules.

## 1. Introduction

The ribosome is the universal ribonucleoprotein machinery that translates the genetic message stored in the mRNA into proteins. Its function is highly conserved and tightly regulated in all domains of life. The ribosome is composed of two subunits, one large ribosomal subunit (LSU; 50S in bacteria and 60S in eukaryotes, depending on their sedimentation coefficient (Svedberg unit)) and a small ribosomal subunit (SSU; 30S in bacteria and 40S in eukaryotes). However, its composition varies remarkably between prokaryotes and eukaryotes. Ribosomes in bacteria are constituted by three types of ribosomal RNA (rRNA) and 54 individual ribosomal proteins (RPs), reaching a molecular weight of approximately 2.5 MDa. In eukaryotes, instead, ribosomes exist as much more complex machineries. In the model organism *Saccharomyces cerevisiae*, for instance, the ribosome is made of four types of rRNA and 79 individual RPs, reaching a molecular weight of 3.3 MDa. Further compositional differences exist between higher and lower eukaryotes, with mammalian ribosomes containing an additional RP and, more importantly, 1 MDa of rRNA extensions called expansions segments (ESs) [1,2]. During the last two decades, several research groups have contributed with groundbreaking studies to unveiling the intimate structure of the ribosome in different organisms and shedding light on its functioning during protein synthesis. At the beginning of this very exciting period of time, X-ray crystallography was the principal method used to determine the structure of the entire bacterial ribosome and its individual ribosomal subunits, as exemplified by several works [3,4,5,6,7,8,9]. Although the earliest low-resolution structure of the *E. coli* 70S ribosome was visualised by cryo-EM [10], X-ray crystallography has been for a long time the only technique yielding high-resolution electron density maps that could be used to build atomic models. Thus, the first advances in ribosome crystallography enabled describing, at atomic level, the interactions of the translating ribosome with its main functional ligands such as transfer RNA (tRNA) and mRNA in bacteria [4,11,12]. In 2011, another major breakthrough in the field was achieved by Yusupov’s group, who determined the crystal structure of the eukaryotic 80S ribosome from *S. cerevisiae* at 3.0 Å resolution [13]. The resulting model allowed the identification of a functional “core” that is conserved with prokaryotes and is responsible for tRNA and mRNA binding, mRNA decoding and peptidyl transferase reaction [1,13]. The larger number of RPs and the additional ESs are mainly restricted to peripheral regions of the eukaryotic ribosomes and suggested to be exclusively involved in regulation of ribosomal functions [1].

The protein synthesis cycle in all cells is characterised by three main phases: (1) initiation, where the 5′ end of the mRNA is scanned by the SSU until the initiation codon (AUG) is recognised; (2) elongation, where LSU joins the SSU to yield a translation competent ribosome. At this step, decoding of mRNA drives the sequential addition of amino acids carried by the tRNAs, and peptide bond formation occurs in the peptidyl transferase centre (PTC); (3) termination and recycling, where the ribosome encounters a stop codon and triggers release of the nascent polypeptide chains and subunit splitting (Figure 1) [1].

The key role played by the ribosome within the cell has made it one of the favourite targets of antibiotics in bacteria (reviewed in [14,15,16,17,18]). These small molecules, synthesised by fungi or soil bacteria as a self-defence mechanism, interact within the functional sites of the ribosome, thus hindering translation. Extensive studies in the past (for more detail see reviews above), but also more recent ones [19,20,21,22], have shed light on the functional implications of protein synthesis inhibition triggered by the binding of antibiotics and on how bacteria can develop resistance. Different classes of antibiotics have been shown to cluster in the Aminoacyl (A), Peptidyl (P) and Exit (E) binding sites within the 50S and 30S subunits [14,17]. Additionally, antimicrobial compounds have been identified to tether to the SSU either to prevent the binding or, very surprisingly, to stall the movement of mRNA during elongation [15]. Structural works focused on unravelling the binding modes of these molecules within the ribosome have been, and yet are, of paramount importance for understanding the molecular basis of inhibition in bacteria and guiding drug design strategies to develop novel antibiotics able to elude resistance mechanisms [23].

Intriguingly, but not totally surprisingly, the eukaryotic ribosome is also the target of naturally derived small molecule inhibitors (Figure 1 and Figure 2, Appendix A). Several compounds are known to hinder translation at different steps (Figure 1), as demonstrated by extensive functional works [24,25,26,27,28,29,30] (and reviewed in [31,32,33]). But until recently, no atomic-resolution information on the mode of action of ribosome inhibitors in eukaryotes was available. The later advances in crystal structure determination of 80S ribosomes from the model organism *S. cerevisiae* allowed applying X-ray crystallography to solve the first high-resolution structures of inhibitors in complex with the eukaryotic ribosome. In this first study, the molecular details of binding of 12 eukaryotic-specific and 4 broad-spectrum inhibitors in complex with the yeast 80S ribosome was unveiled [34]. All compounds unambiguously fit within the vacant 80S ribosome and were revealed to cluster at ribosomal functional sites, similarly to bacteria (Figure 1 and Figure 2, Appendix A) [34]. The study defined common principles of targeting and resistance in eukaryotes, provided insights into translation inhibitor mode of action and revealed the structural determinants responsible for species selectivity. Particularly, this structural investigation allowed proving the existence of an exclusive eukaryotic-specific feature—the large ribosomal subunit tRNA E-site as a target of translation inhibitors.

Inhibitors of eukaryotic protein synthesis, besides being routinely used for research purposes, hold great potential as anticancer therapeutics and to correct for genetic disorders [31,35]. A striking example is represented by the drug omacetaxine mepesuccinate (also called homoharringtonine (HHT)) used for treatment of chronic myeloid leukaemia (CML) (Figure 1 and Figure 2). HHT, marketed as Synribo™, is the first protein synthesis inhibitor approved by the Food and Drug Administration to be prescribed to individuals that develop resistance towards tyrosine kinase inhibitors, which are commonly used as first line of CML treatment [36,37]. The majority of cancers are caused by a hyper-proliferative phenotype that is responsible for an indefinite growth. Fine-tuning the activity of translation inhibitors would likely represent a promising way to counteract the rapid growth of cancer cells, but thorough optimisation is needed to avoid secondary effects. Furthermore, the use of ribosome inhibitors to treat human diseases has also received particular attention to promote premature termination codon (PTeC) read-through in rare genetic disorders [38,39,40].

In addition, there is increasing evidence that rRNA modifications and pervasive sequence variation in rRNA can be related to differential expression in human cells and tissues to originate “specialised” ribosomes, suggesting also a possible link with cancer onset [41,42,43,44]. Different patterns of rRNA modifications expand further the complexity of designing more potent and highly specific anticancer drugs. Visualising these “specialised” ribosomes will provide new understanding at functional level but, more importantly, will serve to gather a much more refined structural interpretation of the ribosomal druggable sites.

We review herein the structural data available for small molecule inhibitors that target the cytosolic eukaryotic 80S ribosome. Several of these structures were obtained by X-ray crystallography using the yeast 80S ribosome, given its high level of conservation with mammals. Nonetheless, there are now a few examples available of high-resolution cryo-EM reconstructions that have identified and positioned unambiguously inhibitors within the human 80S ribosome. We conclude this review by briefly discussing the potential and limitations of both techniques to achieve the needed information to deliver more potent and specific anticancer drugs that could enter clinical trials in the near future.

## 2. Inhibitors Targeting the Large Ribosomal Subunit

The peptidyl transferase centre is the region within the LSU where peptide bond formation occurs. It requires the aminoacyl-tRNA and peptidyl-tRNA to be properly accommodated in the A- and P-sites, respectively; correct positioning of the two tRNAs promotes nucleophilic attack of the amino-terminal of the amino acid carried by the A-site tRNA towards the carbonyl group of the elongating nascent chain [45,46]. Upon peptide bond formation, the new polypeptide chain elongates and finally protrudes out from the peptide exit tunnel (PET), where it encounters dedicated chaperones that promote its correct folding [47]. Once discharged, the deacylated tRNA moves further towards the E-site and, through concerted movements of the L1-stalk, is allowed to exit the ribosome.

Structural studies in bacteria and eukaryotes have confirmed that one of the best strategies to halt translation is by interfering with all these steps during elongation (Figure 1 and Figure 2 and reviewed in [14,17,18]). The PTC is one of the major binding sites for inhibitors targeting the LSU, especially for members of the alkaloid and mycotoxin families (Figure 3) [14,34,48,49]. It was known that the PET is targeted by a large family of compounds in bacteria, the macrolides; however, recent studies have demonstrated that this region can also be targeted in eukaryotes (Figure 4) [16,20,50]. Finally, the achievement of high-resolution structures of the yeast and human ribosomes allowed unveiling the binding modes of inhibitors targeting the E-site of the LSU, an exclusive eukaryotic-specific feature (Figure 5) [34,51,52].

Very interestingly, specific translation elongation inhibitors, such as anisomycin (ANI), lactimidomycin (LTM) and deoxynivalenol, trigger significant activation of the JNK/p38 MAPK signalling pathway, leading to rRNA cleavage and cell death (reviewed in [31]). Importantly, a recent work has demonstrated that this process, called a ribotoxic stress response (RSR), is initiated by the formation of a few colliding ribosomes at intermediate concentrations of inhibitor (in the specific case, ANI) [53] (and reviewed in [31]).

### 2.1. Drugs Binding to the 60S LSU Peptidyl Transferase Centre A-Site

Inhibitors targeting the LSU tRNA A-site, upon binding, induce structural rearrangements in their proximity that can propagate up to 15 Å away from the PTC [2,34]. Once bound, the compounds impair entrance of the aminoacyl-tRNA into the A-site, through a mechanism of steric hindrance with the amino acid moiety of the charged tRNA (Figure 3a).

Alkaloids are a large class of compounds that target the A-site cleft of the PTC in both bacteria and eukaryotes [14,17,34]. Despite the overall ability to bind to the same region on the LSU, compounds within the same family can adopt distinct poses into the A-site pocket, establishing peculiar sets of interactions (Figure 3a). The natural plant alkaloid HHT is of particular interest given its approval to treat a particular type of blood cancer. The crystal structure of the yeast 80S ribosome in complex with HHT revealed that its cephalotaxine scaffold, which differs from the commonly found methylenedioxy-phenanthridine skeleton in other alkaloids, establishes contacts with the 25S rRNA, in particular with the conserved residues A2820 and C2821. HHT possesses an ester substitution that points towards the P-site, creating an electrostatic interaction with the “flipped-up” conserved rRNA residue U2875 (Figure 3b). In the same work the X-ray structures of two *Amaryllidaceae* alkaloids (AAs), lycorine (LYC) and narciclasine (NAR), were disclosed. AAs are natural compounds that can be extracted from *Amaryllidaceae* bulbs, which share a common chemical architecture and have been shown to inhibit cancer cell growth by halting protein synthesis [24,29,49,54]. A remarkable finding is the position of the dioxole moiety with regard to the A-site cleft. While for HHT and NAR it faces rRNA residue C2821 to establish π-stacking interactions between its aromatic ring and the nucleobase, in the case of LYC, the main scaffold is rotated 90 degrees inward to create a new pattern of hydrogen bonds instead (Figure 3a). More recently, another natural AA product has gathered particular interest for its anticancer properties: haemanthamine (HAE, Figure 3c). The crystal structure of HAE in complex with the yeast 80S ribosome unveiled its binding mode, which is very similar to that of the closely related NAR. The two compounds induce a displacement of A2820 and C2821 toward the core to allow the methylenedioxy-phenanthridine moiety to establish stacking interactions with the rRNA residue C2821. Surprisingly, in the case of HAE, there is a conformational rearrangement of a conserved rRNA residue, U2875, which adopts an extended “flip-up” position, similarly to ANI (Figure 3d) and agelastatin A (AglA), upon binding of the inhibitor and promotes the formation of π-stacking interactions with the aromatic ring of the HAE (Figure 3e). Remarkably, intermediate rearrangements of U2875, when compared to the vacant 80S, have been also observed with the other alkaloids targeting this pocket, which points to a stabilising role of such residue in the accommodation of the drugs (Figure 3b–e). In contrast to most of the compounds tested in the work of Garreau de Loubresse and colleagues, AglA function was fully characterised only recently [48]. AglA represses cancer cell growth, and biochemical studies have shown that it does so by halting the elongation step of translation. The crystal structure of this alkaloid in complex with the 80S ribosome unveiled its binding site to be the A-site cleft of the LSU, in agreement with foot-printing data [48]. In case of AglA, however, the drug establishes an unusual halogen–π interaction through its bromine with the nucleobase of U2875. Additionally, the rRNA residue A2820 slightly rotates towards the B ring of the drug to form another π–π stacking interaction (Figure 3e), locking further AglA into the pocket. Interestingly, both for HAE and AglA, accommodation within the A-site cleft promotes a partial outward rotation of the U2873 (U2504, bacterial numbering and kept in parenthesis herein) nucleobase, which in turn promotes additional solvation, through the binding of an ion or water molecule, in proximity of the drugs [34,49]. Members of the mycotoxin family, such as verrucarin A, deoxynivalenol and T2-toxin, have been shown to bind within the same A-site cleft and sit in a very similar way as the previously described inhibitors (Figure 2 and Figure 3). The same goes for members of the alkaloid family, which promote the conformational rearrangement of U2875, although not as dramatically as HAE and AglA, to establish further compound-specific interactions.

The specificity of these inhibitors towards eukaryotes likely rises from the very minor, yet indispensable, differences in sequence with bacterial ribosomes [1,34]. For instance, yeast A2453 is replaced with U2822 in bacteria, likely preventing the displacement observed in the HAE/80S structure compared with the vacant yeast 80S ribosome. This sequence variation would not allow the conserved rRNA residue U2504 to adopt a conformation similar to that of the yeast ribosome, and as a consequence, it would sterically clash with HAE, impeding its binding to bacterial ribosome. Similarly, A2397 of the yeast 25S rRNA is replaced by C2055 in bacteria, which is also suggested to influence the conformation of U2873. Thus, the rRNA sequence divergence in these proximal positions likely accounts for the discrimination of the binding of either bacterial-specific or eukaryotic-specific inhibitors.

### 2.2. Drugs Binding to the Peptide Exit Tunnel

The remarkable finding that macrolides do not stop global translation but inhibit the synthesis of a subset of proteins paved the way to understanding the molecular mechanism of function of drugs binding within the PET (reviewed in [16]). A recent study identified a compound, named PF846, that specifically inhibits translation of the proprotein convertase subtilisin/kexin type 9 (PCSK9) by stalling the actively translating human 80S ribosomes soon after encountering the PCSK9 signal sequence (Figure 4a,b) [50,55,56,57,58].

The authors hypothesised that PF846, upon recognition and successive alteration of a particular structural motif present in the nascent chain (NC) of specific proteins, promotes stalling of the ribosome in a conformational state that would impair the function of the GTPase elongation factor 2 (eEF2) and the related mRNA and tRNA translocation to the next codon [50]. Such conformational rearrangements in the PTC would also strongly affect the release of the nascent chain, thereby preventing translation termination to occur. It is very interesting to note the peculiar halogen–π stacking interaction that the chlorine atom of PF846 establishes with the universally conserved A1588 (A752) of 28S rRNA, a rather unusual chemical bonding that recently emerged as a promising method of drug functionalisation [48,51,52,57] (Figure 4b). The aromatic ring where the halogen atom sits additionally forms π–π stacking with the universally conserved U4525 (U2609) (Figure 4b). Surprisingly, only the pyridine ring of PF846 is suggested to interact with the NC; however, density for this moiety is worse than the rest of the compound, probably because of the increased mobility in this region and the high variability of the stalled NC itself. Despite having a number of contacts conserved between eukaryotes and prokaryotes, the 6-member cyclic ring of PF846 would sterically clash with the bacterial *N1*-methyl-G745, which in humans is C1581 and is conserved in eukaryotes in general. Additionally, G1582 of the 28S rRNA in the human LSU is replaced by A746 of the 23S rRNA in bacteria, which adopts a position that is incompatible with binding of the drug. These distinctions in nucleotide identity and positioning provide a rationale for PF846′s specificity for eukaryotic ribosomes [57].

### 2.3. Drugs Binding to the 60S LSU Peptidyl Transferase Centre P-Site

The broad-spectrum protein synthesis inhibitor blasticidin S (BlaS), in contrast to all the previously described compounds, binds to the LSU tRNA P-site of PTC, in proximity to important residues that drive peptide bond formation (Figure 4a–d) [14,34,59,60]. The BlaS binding mode within the PTC P-site is very well conserved between eukaryotes and prokaryotes, and thus, it is suggested to be its mechanism of function. In eukaryotes, this binding pocket is not targeted by other known inhibitors, as it is instead the case in bacteria, where a new class of compounds, the group A streptogramins, has been demonstrated to occupy both the A-site cleft and the P-site within the PTC [17].

In the vacant yeast 80S, BlaS pyrimidine moiety establishes Watson–Crick base pairing with the facing G2619, while the β-arginine peptide moiety directly contacts the rRNA backbone of residues A2969 and C2970 of the PTC (Figure 4c) [34]. However, recent cryo-EM reconstructions of in vitro reconstituted pre-termination 80S complexes in the presence of BlaS unveiled the mechanism of inhibition in Mammalia. The reconstructions evidenced that BlaS in higher eukaryotes induces a greater displacement of the CCA-tail of the tRNA, the universally conserved sequence at the 3′-end of the acceptor stem where each amino acid is attached, the P-site when compared to bacteria (PDB: 7NWI, [59,60]) (Figure 4d). This likely provokes a series of steric clashes with the incoming substrates into the A-site, such as the aminoacyl-tRNA (during elongation) and eRF1 (during termination), thus hindering both steps of translation. Structural analysis of the BlaS binding mode revealed that the pyrimidine moiety is not forming classic Watson–Crick base pairing, while nevertheless maintaining hydrogen bonding with G2619. This moiety is instead tilted in a way that favours π-stacking interactions between nucleobases with A76 of the P-site tRNA [59].

### 2.4. Drugs Binding to the 60S LSU tRNA E-Site

Cycloheximide (CHX) is a protein synthesis inhibitor mostly used for research purposes, as its high cytotoxicity in human cells has prevented its employment as a potential anticancer drug. It was first identified by foot-printing analysis together with LTM and was found to protect nucleotides located within the eukaryotic E-site in HEK293 cells [28]. The work of Klinge and colleagues using 60S LSU from *Tetrahymena thermophila* initially attempted to visualise CHX in complex with the eukaryotic ribosome, but the quality of the electron density map did not allow placing the inhibitor unambiguously [61]. Three years later, the work of Garreau de Loubresse and colleagues succeeded in solving the structures of CHX, LTM and phyllanthoside (PHY) in complex with the 80S ribosome at high resolution [34] (Figure 2 and Figure 5a–d).

CHX and LTM share a common glutarimide moiety, which establishes highly similar interactions exclusively with 25S rRNA residues. A set of electrostatic contacts with the backbone of G92 and C93 and the nucleobase of U2763, with the aid of a Mg^2+^ atom, likely stabilise this moiety into the pocket, a feature that appears to be conserved for all drugs binding within the E-site (Figure 5b). Additionally, a recent cryo-EM reconstruction has confirmed the mode of binding of CHX onto the human ribosome to be highly conserved [34,62]. LTM, as a member of macrolides, contains an additional 12-member lactone ring that plays a crucial role in binding affinity of the compound (Figure 5c). Although chemically unrelated to glutarimides, PHY binds within the E-site by interacting with the same rRNA nucleotides and, additionally, with the backbone of the eukaryotic-specific eL42 (Figure 5d) [34]. Given its position within the ribosome, CHX is suggested to induce release of tRNA, and conversely, once bound, it prevents tRNA translocation from the P- to the E-site during elongation. The steric hindrance effect of CHX towards the deacylated CCA-end of the E-site tRNA is reminiscent of another chemically unrelated compound, mycalamide A, a natural product with potent antitumour activities that was crystallised, and the structure solved, in complex with the archaeal *Haloarcula marismortui* 50S subunit [63].

In 2006, Robert and colleagues identified a new class of protein synthesis inhibitors, the lissoclimides. The naturally occurring chlorolissoclimide (CL) shares chemical similarities to CHX, since it contains a succinimide moiety that closely resembles glutarimide [64]. A newly established chemical synthesis route permitted obtaining CL and several of its congeners for structural and biochemical characterisation with the eukaryotic ribosome [52]. The crystal structure of the CL/80S complex allowed interpreting the molecular details of the interaction of lissoclimides within the E-site binding pocket. Structural analysis revealed that the imide-containing moiety oriented similarly to those in CHX and LTM, establishing a conserved network of interactions with the 25S rRNA (Figure 5e). Very interestingly, CL was found to additionally interact with other rRNA elements (A2802) and with the backbone of the eukaryotic-specific eL42, similarly to PHY [34,52]. However, one of the most interesting findings was the formation of an unusual halogen–π stacking interaction with a nucleobase of the 25S rRNA (Figure 5e). This kind of interaction has been described (and additionally discussed in the present review) only recently for eukaryotic-specific ribosome inhibitors, such as the drugs AglA [48] and PF846 [57], while it was already reported a decade ago for the bacterial-specific macrolide CEM-101 [65]. Accessing a wide range of congeners of lissoclimides is key to further investigating the potential of the halogen–π interaction, with the aim to generate more potent and specific drugs. Based on in vitro translational inhibition and IC_50_ data [52], the second most potent lissoclimide tested was C45, which contains an additional halogen group in its decalin ring [51]. The crystal structure of the C45/80S complex was of paramount importance to validate previous ab initio docking studies, which suggested a change in conformation of the compound to accommodate into the E-site pocket (Figure 5f). Upon adoption of a twist-boat conformation, C45 reorients its chloride atoms to establish two halogen–π interactions with consecutive nucleobases (G2793 and G2794) (Figure 5f). In the same work, the authors compared binding affinities for members of the lissoclimide families to yeast and human translating complexes, with the aim to further characterise the impact of this unusual RNA–ligand interaction. They concluded that the presence of an additional halogen–π interaction may compensate the loss of other types of bondings and, therefore, represent a valuable alternative to provide further stabilisation of the drug within its pocket [51]. Other strategies, together with exploiting the neighbouring eL42 to establish new contacts, would include maintaining CL or CHX scaffolds to design new functional groups that would mimic the A76 of the CCA-end of the deacylated tRNA, which intercalates between G2793 and G2794 during actual translation [51,52,62].

In bacteria, the E-site pocket is differentially structured and composed exclusively of rRNA (Figure 5g). Despite the relatively high conservation in the region, there are nucleotide differences that act as a barrier, causing resistance towards the binding of compounds to the E-site pocket in bacteria. Importantly, G2794 (yeast numbering) and the facing U2763 are replaced with A2422 and A2393 in bacteria, respectively. These nucleotides, together with U2431 and A2432, form a loop that resides in the same region that in eukaryotes is occupied by eL42, and they are likely responsible for narrowing the pocket in bacteria. In fact, when superposed with yeast 25S rRNA, we can appreciate how, as a consequence of the different rRNA tertiary folding, the universally conserved C2394 (2764, yeast numbering) would sterically clash with the glutarimide moiety of the E-site binding inhibitors (Figure 5g). Moreover, the rRNA backbone of the loop connecting H74 with H88 (conserved in eukaryotes) adopts a remarkably different position and in bacteria would also sterically clash with any incoming drug.

## 3. Inhibitors Targeting the 40S Small Ribosomal Subunit

The SSU is responsible for scanning the mRNA to initiate translation. Additionally, it ensures the accurate decoding of the genetic code during the elongation phase. There are two main regions where inhibitors have been identified to bind to: the mRNA path and the decoding centre (DC) (Figure 2, Figure 6 and Figure 7).

### 3.1. Drugs Binding to the mRNA Path

In the work of Garreau de Loubresse and colleagues, the structures of cryptopleurine (CRY), pactamycin (PAC) and edeine (EDE) with the 80S ribosome revealed that these drugs bind exclusively to the 18S rRNA, and more specifically to a conserved region on the tip of h23 within the 40S E-site [34] (Figure 6a). h23 is known to be involved in interaction with the mRNA and tRNA during elongation; however, the molecular basis of the inhibition triggered by these drugs remains to be fully explained. Biochemical studies have shown that the broad-spectrum inhibitor PAC and the eukaryotic-specific CRY both interfere with the translocation process (Figure 6a) [30,66,67,68]. The C-terminus of uS11 in the yeast ribosome extends towards the binding site of CRY and PAC, likely influencing the binding of the inhibitors. In bacteria the C-terminus of uS11 folds instead towards the core of the SSU; this finding, however, does not fully explain the eukaryotic specificity of CRY with respect to PAC. The broad-spectrum inhibitor EDE displays a different pattern of interactions that are organism dependent. In bacteria, EDE is found to span between the P- and E-sites of the 30S subunit and interfere with binding of the initiator tRNA during translation initiation [66,69]. However, when crystallised in complex with the yeast 80S, EDE establishes interactions with rRNA residues within the mRNA path exclusively limited to the E-site (Figure 6b). Biochemical data have suggested that binding of EDE promotes continuous scanning and hinders subunit association, thus confirming that this compound indeed impairs translation initiation but suggesting that the mechanism could be different [70].

Amicoumacin A (AMA) is another compound interacting mostly with 18S rRNA residues in h23–24 and shown to hinder cancer cell growth [71]. As for EDE and PAC, AMA is a broad-spectrum inhibitor of protein synthesis, and structural comparison of the yeast 80S structure with the bacterial 70S in complex with the drug evidenced the similarity of the interactions established (Figure 6c) [71,72]. Nonetheless, differences in the binding mode of AMA can be ascribed to two universally conserved ribosomal proteins, uS11 and uS7. The former contains a eukaryote-specific extension at the C-terminus that permits establishing direct contacts with the drug. Functional data have demonstrated that mutations within this region confer resistance to CRY, which binds in a similar manner as AMA (Figure 6c); therefore, it is likely that these perturbations might preclude the entrance of AMA. In contrast, the loop of uS7 approaches AMA in the case of bacteria, likely establishing a water-mediated bonding with the drug, while being at a farther distance in eukaryotes [71,72].

### 3.2. Drugs Binding to the Decoding Centre

The DC is the geometrically restricted ribosomal pocket responsible for ensuring the accuracy of translation through inspection of the correct Watson–Crick pairing of the codon of mRNA with the anticodon of tRNA to allow accommodation of the cognate aminoacyl-tRNA into the SSU A-site. Once more, the central role in translation played by the DC makes it a major target of protein synthesis inhibitors, with members of the aminoglycosides as a well-known example (Figure 1 and Figure 2 and Appendix A) (reviewed in [73]). In bacteria, aminoglycoside antibiotics bind with high affinity to the DC, altering translation accuracy at both “sense” and “stop” codons, resulting in extensive translational misreading; additionally, they inhibit tRNA translocation by perturbing the conformation of the highly conserved DC nucleotides [74,75,76,77]. The canonical aminoglycoside-binding site, both in eukaryotes and prokaryotes, is located within the internal loop of helix 44 (h44) of the 18S rRNA (16S rRNA in prokaryotes), which contains the essential and universally conserved nucleotides A1755 and A1756 (A1492 and A1493) (Figure 7a).

In close proximity to these two adenines, evolution has diverged, and as a result, positions G1645 and A1754 in yeast (conserved also in humans) are replaced into A1408 and G1491, respectively, in bacteria. Likewise, the presence of an adenosine instead of guanosine at position 1754 would disrupt Watson–Crick base pairing with C1646 (C1409), an essential structural determinant for aminoglycoside binding [34]. These key differences are likely crucial for accommodation of aminoglycosides within the DC, thus providing a rationale for the discrepancy in binding affinity between domains of life. However, despite the lower affinity for the eukaryotic ribosome, a subset of aminoglycosides has been shown to promote misincorporation of near-cognate aminoacyl-tRNAs at premature termination codons (PteCs) [38,78,79,80,81]. PteCs are inserted into the gene-coding sequence by nonsense mutations, which have been identified in approximately 10% of inherited genetic disorders in humans [40]. Ribosomes, upon encountering a PTeC, terminate translation yielding truncated peptides, which are typically nonfunctional [82]. Therefore, treatments designed to force the ribosome to “read-through” a PTeC and continue translation to restore full-length protein levels could mitigate the impact of nonsense mutations. Several works have shown that aminoglycosides can suppress termination at PTeCs within a number of mRNAs and restore levels of functional protein in mammalian cells (reviewed in [39,82]). However, the drawback of using aminoglycosides is their high toxicity, which can cause hearing loss and kidney damage as potential side effects [38,78,82]. This toxicity does not appear to be attributable to the ability of aminoglycosides to suppress PTCs, but primarily due to their association with off-target sites such as lysosomal membranes and, given their similarities to bacterial ribosomes, the decoding sites of mitochondrial ribosomes [78,79]. Nonetheless, the considerable side effects of aminoglycosides have not hampered the enrolment of clinical trials for treatment of cystic fibrosis and Duchenne muscular dystrophy, as well as dystrophic epidermolysis bullosa, Werner syndrome and specific cancers [38,79,83].

### 3.3. Aminoglycosides Binding to the Eukaryotic Decoding Centre

As mentioned earlier, aminoglycosides generally bind with low affinity to eukaryotic ribosomes. After thorough optimisation of crystallisation conditions to achieve high resolution diffraction data in absence of osmium hexamine, Prokhorova and colleagues obtained four crystal structures of aminoglycosides and unveiled their molecular details of binding: G418 (geneticin), paromomycin (PAR), gentamicin (GENT) and TC007 (Figure 7a–d) [74]. The authors solved the structure of the 4,6-linked aminoglycoside G418 bound to the *S. cerevisiae* 80S ribosome and reported it to exhibit canonical binding to the h44 decoding site, as was shown previously in presence of osmium hexamine (Figure 1 and Figure 2) [34]. G418 ring I directly interacts with the eukaryote-specific residues G1645 and A1754 (A1408 and G1491, respectively) and promotes “flip-out” of the conserved nucleotides A1755 and A1756 from the internal loop of h44 [74].

The structure of the 4,5-linked aminoglycoside PAR in complex with the yeast 80S revealed that rings I and II adopt a position globally similar to those of G418 (Figure 7b). However, when compared with the bacterial PAR/70S structure, ring III is rotated, likely to avoid steric clash with the eukaryotic-specific A1754, while ring IV reorients within the major groove of h44 [74]. Very interestingly, ring I of PAR was shown to establish a hydrogen bond with lysine in position 3 of the eukaryote-specific protein eS30, which approaches h44 from the minor groove. The authors claimed that this interaction, which was not previously observed for G418 when bound to the ribosome, may compensate for the reduced affinity within the h44 decoding site, in order to maintain low levels of miscoding [74].

GENT is the most used aminoglycoside for nonsense suppression studies. GENT is a 4,6-linked aminoglycoside that has been shown to rescue, although for a short period, functional protein levels in mouse models [72,80,84]. The crystal structure of the GENT/80S complex revealed that GENT binds to the canonical binding site within h44 but with a unique mode of interaction (Figure 7c). Surprisingly, ring I of GENT does not face the conserved residue A1754, and nucleotides A1755–A1756 adopt an intermediate “flip-out” configuration relative to the axis of h44 (Figure 7c). Finally, rings I–III of GENT establish additional contacts with U1758 and G1642 (U1495 and G1405, respectively).

Spinal muscular atrophy (SMA) is a leading genetic cause of infantile death, associated with loss of full-length SMN protein [85,86]. Early screenings for drug-induced suppression therapies identified TC007, a three-ring 4,5-linked aminoglycoside, as a promising hit [86]. Tests in SMA mouse models displayed prolonged lifespan while showing tolerable toxicity profiles [85]. Prokhorova and colleagues solved the crystal structure of TC007 in complex with the yeast 80S ribosome and surprisingly unveiled that its main binding site is not within the decoding site of the small subunit (Figure 7d). The authors identified the aminoglycoside to sit above h44, in a pocket close to the intersubunit bridge formed together with H71 from the LSU (Figure 7d). However, in the same work, the structure of the bacterial 70S in complex with TC007 revealed that the aminoglycoside targets the canonical binding site within h44, with rings I and II adopting a similar arrangement as with PAR (Figure 7d) [74]. The authors suggested that differences in TC007 binding to the cytosolic 80S ribosome are likely due to the U2264 and C2265 substitutions (G1921 and G1922, respectively), which might clash with ring II of the drug.

Intriguingly, in the same study, the authors identified several other secondary binding sites for all the aminoglycosides tested. They hypothesised, and confirmed by elegant single-molecule experiments, that the inhibitors might play an active role in the dynamics of subunit rotation as an alternative mechanism of inhibition [74]. In fact, other works in the past have documented that aminoglycosides can bind within the major groove of Helix 69 (H69) of the LSU, involved in a critical intersubunit bridge that directly contacts the h44 decoding site [76,77,87].

Hygromycin B (HygB) is yet another aminoglycoside that is largely used in the scope of research and that has been shown to bind within the DC. This broad-spectrum inhibitor was initially crystallised in complex with bacterial 30S and 70S, while recently visualised within the human SSU by cryo-EM [88,89,90]. HygB has been reported to bind with two different orientations: in the *T. thermophilus* 30S structure, HygB does not induce “flip-out” of the decoding conserved residues [89]. However, when crystallised in complex with the *E. coli* 70S, the inhibitor promotes partial rearrangement of the decoding nucleotides A1492 and A1493, with the former adopting an intermediate conformation while the latter extrudes towards the mRNA path [88]. A very similar pattern is observed in mammals, with HygB promoting flip-out of A1825 (A1493), additionally establishing a few contacts with ring III.

## 4. Conclusions and Future Directions

The recent advances in the field of cryo-EM have made it possible to achieve atomic-resolution reconstructions, as recently proven [91,92]. This is currently restricted to few very well-established examples; however, continuous development in EM hardware and single-particle processing software, coupled to higher knowledge of optimisation strategies for grid freezing conditions, might enhance the feasibility of achieving sub-2 Å overall resolution reconstructions, as shown by the work on the *E. coli* 70S ribosome [93]. At such a level of detail, although not attainable for solvent exposed regions and flexible domains, the solvation shells within inhibitors’ binding pockets become interpretable and can further guide drug development through optimisation of chemical functionalisation of existing compounds. Additionally, the preparation of cryo-EM grids is a rapid process and permits working with concentrations of drugs in the low micromolar range. Crystallisation, conversely, is a slow process (ribosome crystals take approximately 3–4 weeks to reach their final size), and unstable compounds can degrade during this time. To overcome this, grown crystals can be soaked into solutions containing the drug of interest. However, because of the highly viscous components at this stage, which could reduce diffusion of small molecules into the ribosome, high concentrations must be used (in some cases up to the millimolar range), thus requiring significant amounts of compounds. On the other hand, despite the hurdle of producing high-quality ribosome crystals, the great benefit is still the relatively medium–high throughput at which compounds can be screened, with results achievable in the order of 36–48 h post-data collection at a synchrotron source. Electron density maps in the range of 2.9–3.5 Å resolution can provide compelling details about the region of binding and the interaction pattern that a drug establishes within the ribosome and could serve to narrow down the search for the optimal inhibitor.

## Figures and Tables

**Figure 1 cancers-13-04392-f001:**
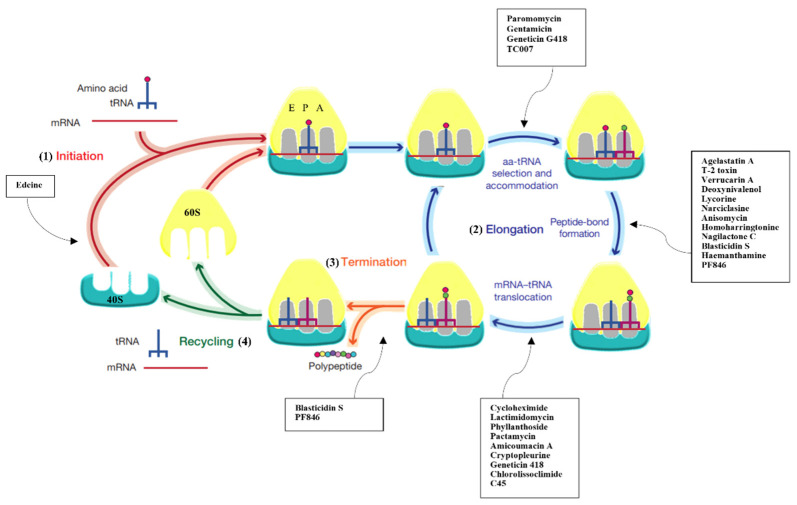
Protein synthesis inhibitors target several steps of translation. Scheme reporting the different phases of protein synthesis: (1) initiation, where initiator tRNA (tRNAi) binds to the 40S subunit upon start AUG codon recognition; (2) elongation, where peptide bond formation occurs upon selection of the correct cognate tRNA; (3) termination, which triggers the nascent chain’s release upon encountering the stop codon; (4) recycling, where 80S splits up to deliver individual subunits back into the cycle. Overview of the pathways that are inhibited upon binding of small molecule drugs presented in this review.

**Figure 2 cancers-13-04392-f002:**
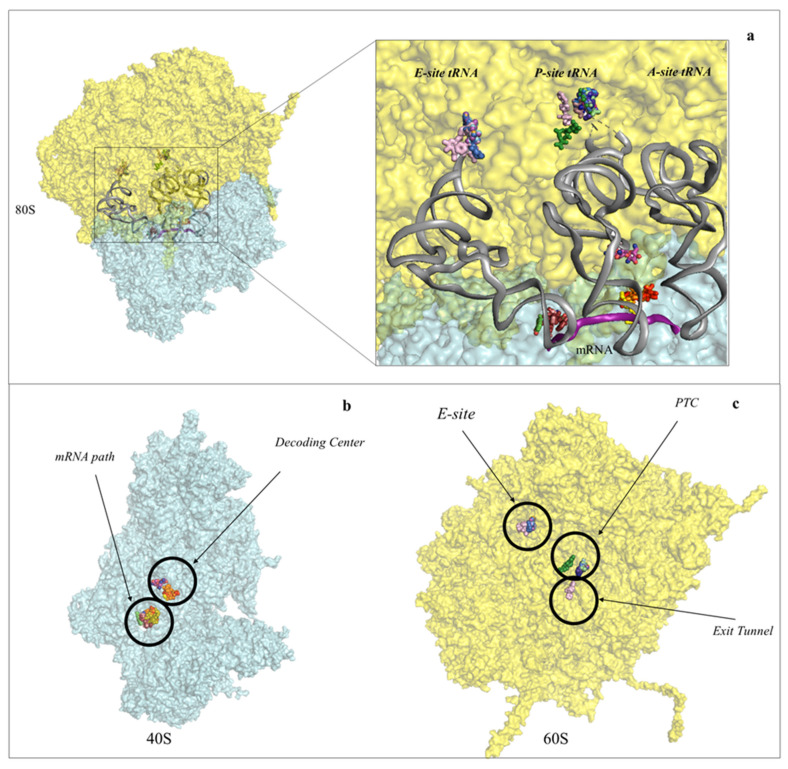
Protein synthesis inhibitors cluster to ribosomal functional sites. (**a**) Left-hand side, overview of the eukaryotic 80S ribosome, shown as surface representation. LSU is coloured in cyan, SSU in pale yellow. The three tRNAs and mRNA were modelled in silico after superposition of the *Thermus thermophilus* elongating complex (PDB: 4V5D) to the yeast vacant 80S model (PDB: 4V88). Right-hand side, zoom-in of the regions where inhibitors bind within the ribosome. For clarity the functional sites are labelled. PTC: peptidyl-transferase centre; PET: peptide exit tunnel; DC: decoding centre; (**b**) binding of small molecule inhibitors to the eukaryotic SSU. Drugs are represented as spheres, SSU as surface. The functional sites targeted are displayed; (**c**) binding of small molecule inhibitors to the eukaryotic LSU. Drugs are represented as spheres, SSU as surface.

**Figure 3 cancers-13-04392-f003:**
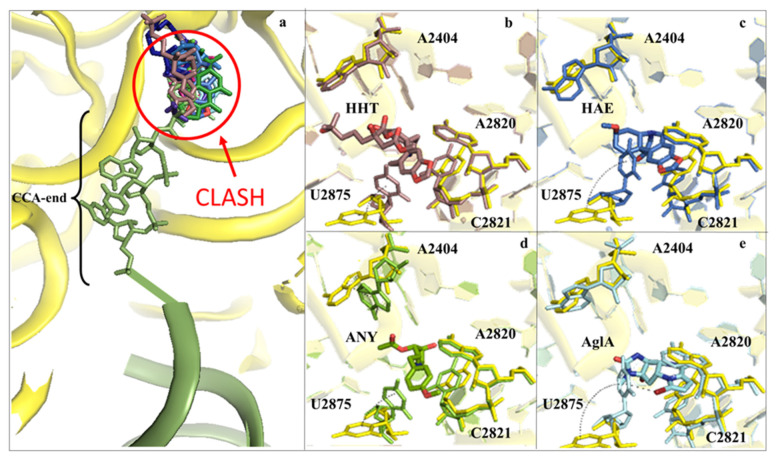
Inhibitors binding to the A-site cleft impede the binding of the aa-tRNA during elongation. (**a**) Overview of the inhibitors that bind to the A-site cleft. All compounds would sterically clash with the incoming charged acylated A-site tRNA, thus impairing accommodation of the following amino acid within the pocket and therefore halting the elongation phase of translation. Anisomycin (ANI) in lemon (PDB: 4U3M), agelastatin A (AglA) in cyan (PDB: 5MEI), deoxyvalenol in fuchsia (PDB: 4U53), haemanthamine (HAE) in blue (PDB: 5ON6), homoharringtonine (HHT) in brown (PDB: 4U4Q), lycorine (LYC) in light blue (PDB: 4U4U), nagilactone C in aquamarine (PDB: 4U52), narciclasine (NAR) in green forest (PDB: 4U51), T2-toxin in purple (PDB: 4U6F) and verrucarin A (PDB: 4U50) in blue navy; (**b**) zoom-in and details of interaction for HHT within the A-site cleft. Proximal rRNA residues are shown as sticks; (**c**) zoom-in and details of interaction for HAE within the A-site cleft; (**d**) zoom-in and details of interaction for ANI within the A-site cleft; (**e**) zoom-in and details of interaction for AglA within the A-site cleft. The rRNA residues that adopt different conformations upon binding of the inhibitors, A2404 and U2875, are shown for clarity. The peculiar halogen–π interaction established between the pyrimidine ring of U2875 and the bromine atom of AglA is also highlighted.

**Figure 4 cancers-13-04392-f004:**
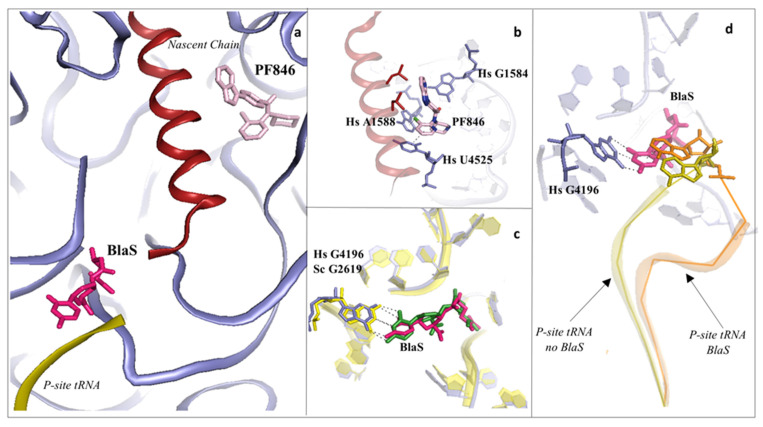
Inhibitors binding to the PTC impair elongation of the nascent chain and can be sequence specific. (**a**) Overview of the PTC and peptide exit tunnel (PET) of the human ribosome. Blasticidin S (BlaS) in green and hot pink (PDB: 4U56 Sc/7NWI Hs) and PF846 in light pink (PDB: 6XA1) bind within the PTC, where the P-site tRNA and nascent chain, respectively, interact with the ribosome; (**b**) PF846 binds to the PET and contacts residues A1588 and U4525 by establishing two different types of interactions, halogen–π and π–π stacking, respectively, each mediated by the chloride atom; (**c**) structural superimposition of human and yeast ribosomes solved in complex with BlaS. The conserved residue G4196 (G2619 yeast numbering) establishes a similar hydrogen bonding network with the compound in both organisms; (**d**) superimposition of translating human ribosome complexes in presence (PDB: 7NWI) and absence of BlaS (PDB: 6XA1). Detail of the displacement of the CCA-end of the P-site tRNA upon binding of the drug, which further promotes an additional π–π stacking with A76, likely preventing the correct accommodation of the tRNA into the PTC. P-site tRNA in absence of BlaS is coloured in olive, while the P-site tRNA in presence of BlaS in orange. Hs: *H. sapiens*.

**Figure 5 cancers-13-04392-f005:**
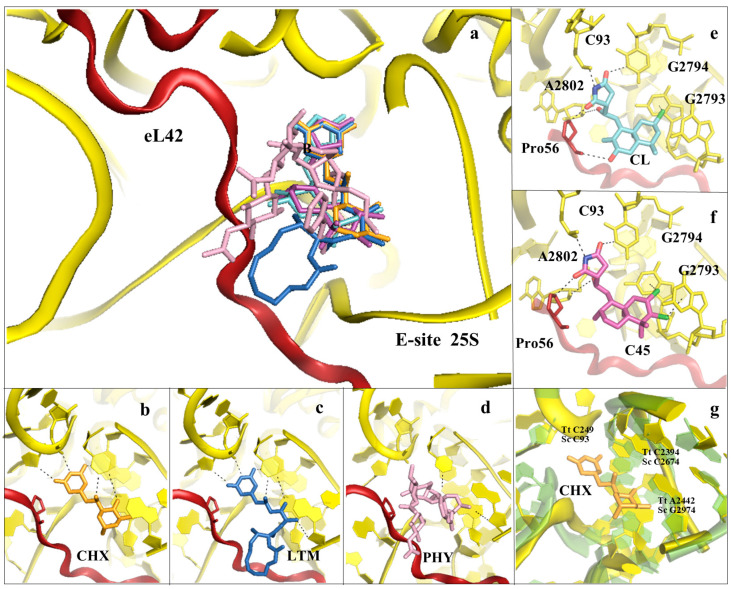
Inhibitors binding to the E-site of the LSU impair the translocation process. (**a**) Overview of the inhibitors targeting specifically the eukaryotic E-site. Cycloheximide (CHX) (PDB: 4U43) in orange, lactimidomycin (LTM) in blue (PDB: 4U4R), phyllantoside (PHY) in pink (PDB: 4U4Z), chlorolissoclimide (CL) in light cyan (PDB: 5TBW) and the synthetic analogue C45 in magenta (PDB: 6HHQ); (**b**) zoom-in and details of interaction for CHX within the E-site binding pocket; (**c**) zoom-in and details of interaction for LTM within the E-site binding pocket; (**d**) zoom-in and details of interaction for PHY within the E-site binding pocket. PHY, in contrast to CHX and LTM, is reported to interact directly with the eukaryotic-specific ribosomal protein eL42. eL42 is coloured in red for clarity; (**e**) zoom-in of CL interaction network within the E-site, with details of the hydrogen bonding and the peculiar halogen–π interaction that lissoclimides establish with rRNA residues. Additionally, CL interacts with the backbone of eL42; (**f**) same view for C45, with highlighted the second halogen–π interaction that the drug establishes with the rRNA. This additional bond might compensate for the lack of the hydrogen bond with eL42, as it occurs with CL. In both images, the type of bonding established between the drug and the ribosome is marked; (**g**) structural superimposition of the *S. cerevisiae* LSU (coloured in yellow) with the LSU of *T. thermophilus* (coloured in light green) (PDB: 4V5D) (rmsd 4.5 Å). Numbers of residues for the two organisms are shown (Sc: *S. cerevisiae*, Tt: *T. thermophilus*).

**Figure 6 cancers-13-04392-f006:**
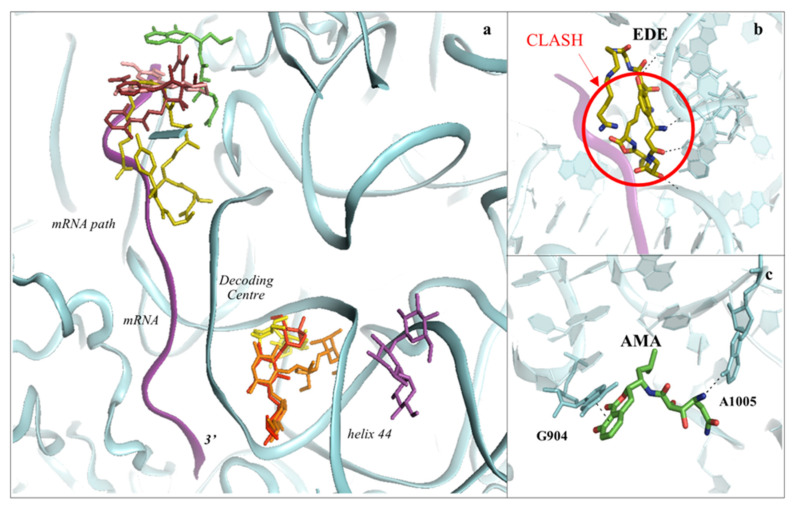
The SSU can be targeted by protein synthesis inhibitors at two main regions. (**a**) Overview of the binding sites for small molecules binding to the 40S subunit. Amicoumacin A (AMA) in green (PDB: 5I4L), cryptopleurine in salmon (PDB: 4U55), edeine (EDE) in olive green (PDB: 4U4N) and pactamycin (PAC) in ruby (PDB: 4U4Y). mRNA is also shown as a ribbon and coloured in purple. Inhibitors have been shown to bind with high specificity to the mRNA path and the decoding centre (DC), thus impairing tRNA–mRNA translocation and altering the translation fidelity process; (**b**) zoom-in and details of interaction for EDE within the mRNA path in correspondence with where the E-site tRNA should interact. The clash with mRNA (shown in red) would prevent the formation of the preinitiation complex; (**c**) zoom-in and details of interaction for AMA within the mRNA path. AMA forms a π–π stacking interaction with the conserved rRNA residue G904 while hydrogen bonding with the conserved A1005 (shown as dashed lines).

**Figure 7 cancers-13-04392-f007:**
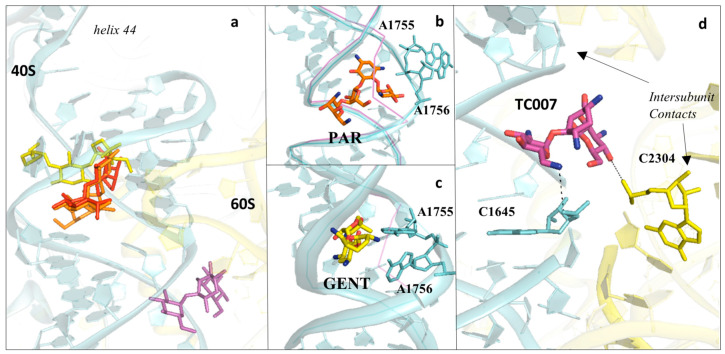
Aminoglycosides bind preferentially to the DC of the eukaryotic ribosome. (**a**) Overview of aminoglycosides binding in correspondence of helix 44 (h44) of the eukaryotic SSU. Paromomycin (PAR) in orange (PDB: 5NDV), geneticin (G418) in red (PDB: 5NDG), gentamycin (GENT) in yellow (PDB: 5OBM), TC007 in violet (PDB: 5NDJ); (**b**) PAR binds within the minor groove of h44 and promotes “flip-out” of the two conserved adenines (A1755/A1756) towards the mRNA channel; (**c**) GENT binds to h44, but docks in a different way compared to PAR, and induces only partial extrusion of the decoding residues. Vacant ribosome (PDB: 4V88) aligned and shown as a purple line for evidencing the conformational change of the minor groove of h44 upon drug binding; (**d**) zoom-in and details of interaction for the aminoglycoside TC007. TC007 directly contacts both LSU and SSU, establishing hydrogen bonds with rRNA residues of the 25S and 18S at a specific intersubunit bridge (B3).

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
