# Peer review of "Inhibition of the Eukaryotic 80S Ribosome as a Potential Anticancer Therapy: A Structural Perspective"

_cancers, 2021, doi:10.3390/cancers13174392_

Round 1

Reviewer 1 Report

This review manuscript from Simone Pellegrino and colleagues is a timely work which incorporates basic and more recent findings on drugs inhibiting the eukaryotic ribosome from a structural perspective.

The review has obvious (but not exclusive) implications for cancer treatment and for the development of cancer therapies directed to the ribosomes of cancer cells.

The work is well organized and I only suggest some minor corrections:

Figure 3 legend "... thus impairing accommodation of the correct amino acid...": this is confusing in my opinion since also the incorporation of work amino acids is equally impaired, I suggest to reword the sentence (e.g. impairing the accommodation of the following amino acid...).

page 7 line 198, HTT need to be expanded at its first use 

page 11 line 357 "the CCA-end of the E-site" can be understood only by people in the field, I suggest to add some brief explanation.

page 12 line 408 Section 3 title looks not entirely appropriate since many drugs are then discussed not only those with an anticancer activity (particularly in section 3.3). Also in coherence with section 3 I suggest not to refer only to anticancer drugs.

Reviewer 2 Report

In my opinion, this is an outstanding review and I would highly recommend publishing it. My only concern is that currently the fair title of this manuscript should be “Inhibition of the eukaryotic 80S ribosome: a structural perspective”. In my opinion, it is not properly described what additional properties of ribosome inhibitors can make it possible to apply it as anti-cancer drugs. Indeed, many ribosome inhibitors are so notoriously poisonous that it is hard to imagine how it can be applied to treat cancer without killing the person. For example, one of the compounds described in this study, verrucarin A, is so toxic that it is fatal upon skin contact. Clearly, there should be some particular properties for ribosome inhibitors apart from direct effect on translation, which would make it a good candidate as an anti-cancer drug. One can speculate that one of such properties may be increased stability of a particular inhibitor inside a cancer cell - this is reasonable as metabolic reprogramming of cancer cells is a very well-known phenomenon.

Another important point which is currently missing in this review is functional consequences of treatment with ribosome inhibitors on cell fate. There are several important publications which describe various scenarios of ribotoxic stress response which can be activated upon treatment with certain ribosome inhibitors. Here I can mention the recent exciting work from Rachel Green’s lab, PMID: 32610081. The authors showed that treatment of cells with low doses of translation inhibitors (anisomycin and emetine) results in ribosome collisions which in turn activate specific signaling pathways through the sensor of collided ribosomes, the Zak kinase. It is reasonable to speculate that treatment with ribosome inhibitors may be beneficial in case if particular cancer cells have dysregulated ribotoxic response pathways.

I understand that the main focus of this review is structural perspective, so the authors may decide not to follow my suggestions. But a brief discussion of these topics may be really helpful for those readers who are not interested in ribosomes but are interested in developing and implementing novel anti-cancer drugs. 
